Comparative proteomics analysis reveals the difference during antler regeneration stage between red deer and sika deer

Su Hang 1
Tang Xiaolei 2
Zhang Xiaocui 2
Liu Li 2
Jing Li 1
Pan Daian 3
Sun Weijie 4
He Huinan 4
Yang Chonghui 4
Zhao Daqing 4
Zhang He 13843148162@163.com 3
Qi Bin qibin88@126.com 2
1 Practice Innovations Center, Changchun University of Chinese Medicine , Changchun , China
2 College of Pharmacy, Changchun University of Chinese Medicine , Changchun , China
3 School of Clinical Medicine, Changchun University of Chinese Medicine , Changchun , China
4 Jilin Ginseng Academy, Changchun University of Chinese Medicine , Changchun , China
Kiss Ibolya
Electronic publication date: 2019 Jul 17
Publication date: 2019
Volume: 7
Electronic Location ID: e7299
Received 2019 Jan 10; Accepted 2019 Jun 14
Copyright: ©2019 Su et al.
Copyright year: 2019
Copyright holder: Su et al.
License: This is an open access article distributed under the terms of the Creative Commons Attribution License, which permits unrestricted use, distribution, reproduction and adaptation in any medium and for any purpose provided that it is properly attributed. For attribution, the original author(s), title, publication source (PeerJ) and either DOI or URL of the article must be cited.
License URL: https://creativecommons.org/licenses/by/4.0/

Keywords: Deer antler, Proteomics, Regeneration, Oxidative phosphorylation, Ribosome, Extracellular matrix interaction, PI3K-Akt

Funding: State Administration of Traditional Chinese Medicine of the People’s Republic of China ZYBZH-Y-JL-26 National Key Research and Development Program of China 2018YFC1706600 This study was funded by the State Administration of Traditional Chinese Medicine of the People’s Republic of China, Grant number ZYBZH-Y-JL-26. This work was also supported by the National Key Research and Development Program of China, Grant number 2018YFC1706600. The funders had no role in study design, data collection and analysis, decision to publish, or preparation of the manuscript.

==============================
Deer antler, as the only mammalian regenerative appendage, provides an optimal model to study regenerative medicine. Antler harvested from red deer or sika deer were mainly study objects used to disclose the mechanism underlying antler regeneration over past decades. A previous study used proteomic technology to reveal the signaling pathways of antler stem cell derived from red deer. Moreover, transcriptome of antler tip from sika deer provide us with the essential genes, which regulated antler development and regeneration. However, antler comparison between red deer and sika deer has not been well studied. In our current study, proteomics were employed to analyze the biological difference of antler regeneration between sika deer and red deer. The proteomics profile was completed by searching the UniProt database, and differentially expressed proteins were identified by bioinformatic software. Thirty-six proteins were highly expressed in red deer antler, while 144 proteins were abundant in sika deer. GO and KEGG analysis revealed that differentially expressed proteins participated in the regulation of several pathways including oxidative phosphorylation, ribosome, extracellular matrix interaction, and PI3K-Akt pathway.

Introduction

Deer antlers are the mammalian regenerative organ, which is covered with abundantly vascular skin called velvet. The antler regeneration initiates in every spring (Fennessy, 1984). At early regeneration stage, the main beam of antler grows dramatically on posterior growth center (Li et al., 2014). Subsequently, at the fast regeneration stage, tines of antler will form on the anterior growth center. During the antler regeneration stage, the growth rate of red antler will reach to 100 g per day within 100 to 120 days (Borsy et al., 2009). The weight of the full size of red deer antler generally reached seven to nine kg (Borsy et al., 2009). However, the sika deer antler with three to six kg is less than red deer (Hu et al., 2019). Following into autumn, the antler gradually reaches the full size and then has calcification accompanied with the loss of velvet and vessels (Li et al., 2014). Over the past decades, this special feature attracted researchers to study on the development mechanism underlying antler regeneration. The antler generation through modified endochondral ossification was first reported by Banks & Newbrey (1983). Antler histogenesis and tissue depletion study provided sufficient evidence that pedicle periosteum drive antler to regeneration (Li et al., 2007; Li & Suttie, 1994; Li, Suttie & Clark, 2004). Besides the pedicle periosteum study, researches about the growing tip of antler with the fastest growth rate are also a hotspot (Deb-Choudhury et al., 2015; Gyurjan et al., 2007; Yao et al., 2012).

Molnar et al. (2007) identified two expression clusters for 36 genes that were mainly expressed in consecutive tissue zones of antler tip of red deer. The first gene expression cluster was found to involve with ribosome pathway and the second cluster could be related with tumor biology. Furthermore, the activated runx2 pathway up-regulated the expression of mineralization proteins in cartilaginous tissue of red deer antler (Steger et al., 2010). Steger et al. (2010) demonstrated that antler regeneration model can be also used to study human osteoporosis. To further deeply study deer antler, the systematic study of sika deer antler tips based on transcriptome provided more information about the key regulated genes related to cartilage development (Yao et al., 2012). Li et al. (2012a) first used proteomic technology to disclose the molecular mechanism of antler stem cells from red deer. This study identified the key markers of antler stem cells, which were transcription factors POU5F1, SOX2, NANOG, and MYC. Moreover, distinctive pathways involved in sika deer antler development were found through the comparative proteomics between potentiated and dormant antler stem cells (Dong et al., 2016). Notably, PI3K-AKT, never identified in red deer antler, played an indispensable role in sika deer antler regeneration. We found that red deer or sika deer was selected as experimental animal to study on antler regeneration in last decade. Also, both of them have the potential to be developed as the model of regenerative medicine. However, the difference of antler regeneration between two deer species still remains unknown. Unfortunately, a previous study showed that the histogenetic aspects of these antlers were indistinguishable between red deer and sika deer (Li, 2013; Li et al., 2014). Except for the size of two antler groups, we never know any other differences between the two antler groups before. In some cases, the term “antler” means that the antler was collected from sika deer or red deer. The antler, as Chinese tradition medicine, is collected from sika deer or red deer according to Chinese Pharmacopeia (Pharmacopoeia Commission of the Ministry of Health of the People’s Republic of China, 2015). We realized that comparative study of antler regeneration between red deer and sika deer has been a “scotoma” all the time. Hence, it is necessary to make the comparative study of these two antler groups. In our current study, the proteomics technology was applied to reveal the differences of the protein level between them.

The aim of our study was to disclose the differentially expressed proteins between two antler groups. Furthermore, we want to remind other researchers that these two antler systems are different. The researchers should notice that whether these different pathways affect their study, when they choose antler to study on regenerative biomedicine. The proteomics based on mass spectrum can identify thousands of proteins, which were processed by bioinformatic software to achieve the protein properties, consisting of protein abundance, protein subcellular distribution, protein functional classification, and correlative biological processes (Larance & Lamond, 2015). Thus, proteomics will accurately work out essentially changed molecular pathways of antler regeneration between red deer and sika deer. The differentially expressed proteins identified in comparative proteomic may facilitate the discovery of novel molecular pathways related with antler growth.

Materials and Methods

Tissue sampling

We randomly chose three male red deer and three male sika deer as study subjects in Chinese local deer farm, which was located in Shuangyang district, Changchun city, Jilin province (N43°33′56.03″, E125°29′23.15″). All procedures of the sample collection were approved by the Animal Ethics Committee of Changchun University of Chinese Medicine. They are all 5-year-old deer with antler regeneration for 80 days after casting previous antlers. Antler tissues were harvested from anaesthetized deer. Briefly, the distal five cm of the antler tip, regarded as the growth center, was removed and sectioned sagittally by the electric saw to collect five mm to eight mm thick tissue section (Li, Suttie & Clark, 2005). Then, the tissue section was cut into small pieces, flash frozen and ground into powder in liquid nitrogen, combined and stored at −80 °C ready for proteomics study.

Protein extraction and FASP digestion

The antler powder samples were extracted with SDT-lysis buffer (4% (w/v) SDS, 100 mM Tris-HCl pH 7.6, 0.1 M DTT) in a sonic dismembrator for 20 min. And then the suspension was incubated at 4 °C for 60 min and subsequently heated at 95 °C for 5 min. The incubation solution was centrifuged at 16,000 g at 4 °C for 15 min. The clarified supernatant was collected to another tube, and then the pellet extracted again in lysis buffer according to the above method. The combination of supernatants was centrifuged at 30,000 g and 4 °C for 15 min. The concentration of proteins was quantified using the bicinchoninic acid assay (BCA assay). 200 µg of protein from each antler sample was digested with trypsin according to filter-aided sample preparation (FASP) protocol proposed by Wisniewski et al. (2009). Subsequently, two µg of tryptic peptides were purified using C18 spin columns (Pierce™ C18, Sigma), and then dried via lyophilization and rehydrated with 40 µl of 0.1% formic acid. The content of the peptide sample was quantified based on colorimetric protein concentration assays (BioRad).

LC-MS/MS analysis

Purified digested peptide fractions were analyzed by nano LC-MS/MS using EASY-nLC 1200 coupled to Q Exactive™ mass spectrometer (Thermo Fisher Scientific). The peptide samples were separated with a 120 min linear gradient from 0% to 55% buffer B (84% acetonitrile, 0.1% Formic acid) on a reverse phase peptide column (Acclaim™ PepMap™ 100 C18, Thermo Fisher Scientific connected) to the EASY capillary C18 column (75μm inner diameter, 10 cm long, three μm resin; Thermo Fisher Scientific) at a flow rate of 300 nl/min. The data-dependent top10 method was used to acquire MS data by choosing the most abundant precursor ions for HCD fragmentation. The full MS survey scans were acquired at a resolution of 70,000, with automatic gain control target 3e6 and a range of m/z 300–1800. HCD spectra scan with a resolution of 17,500 at m/z 200 and the isolation window was two m/z. The dynamic exclusion was 40 s.

Database search and protein quantification

Mass spectra were processed by the peptide search engine Andromeda in MaxQuant software, which searched the data against uniprot_Cervus_20600_20180413.fasta retrieved from UniProt database. The correlated parameters were defined as follows: digest enzyme of trypsin, max missed cleavage of 2, fixed modifications of carbamidomethyl, variable modifications of methionine oxidation and N-terminal acetylation, the precursor mass window in the main search of six ppm and MS/MS tolerance of 20 ppm. The cutoff value of the false discovery rate was defined as 0.01 to identify peptides and proteins. The protein abundance was normalized and calculated according to the algorithms described in Cox et al (Tyanova, Temu & Cox, 2016). Data from the same antler group were merged into the same group, and the LFQ intensities of the proteins were log 2 transformed to obtain log 2 fold-change between two antler groups. To improve the accuracy of comparison, the data were retained with at least two of three valid values (greater than 0) in at least one group, and missing values were replaced by generating random numbers from a Gaussian distribution that well represents the distribution of low-abundance proteins (Qian et al., 2018). If a certain protein has two valid values in one group and only one in the other group, it will also be eliminated. Two-sample T-test is applied for determining if the means of the LFQ intensity values of two groups of deer antlers are significantly different from each other through the analysis of Perseus software (Tyanova et al., 2016). To define differentially expressed proteins (DEPs), fold change thresholds were set at 2.0 or 0.5 with p-value < 0.05. The mass spectrometry data in proteomics study have been submitted to the ProteomeXchange Consortium via PRIDE partner repository, with the dataset identifier PXD012090 (Vizcaino et al., 2016).

Bioinformatic analysis

To study on the biological function of the differentially expressed proteins, the identified proteins were used to run Gene Ontology (GO) analysis. Firstly, the homology search was performed for all identified sequences against the red deer database using localized NCBI BLAST+ software. And then GO annotation and enrichment analysis was completed using BLAST2GO software (Gotz et al., 2008). The result of GO annotation presented that the properties of the identified proteins in three functional categories, which are cellular component (CC), biological process (BP), and molecular function (MF). In addition, all differentially expressed proteins and significantly changed metabolites were searched against the Kyoto Encyclopedia of Genes and Genomes (KEGG) database and obtained the corresponding KEGG pathways (Moriya et al., 2007). To further disclose the mechanism underlying differentially expressed proteins and clarify the functional cluster of differentially expressed proteins, GO enrichment analysis was performed by Fisher’s exact test. Notably, GO terms with p values < 0.05 were regarded as significant enrichment.

Results

Protein identification and differentially expressed proteins (DEPs)

In total, 4,439 unique peptides and 1,060 proteins were identified through MaxQuant software against UniProt database. All identified protein from these two groups are listed in Table S1. An obvious difference in the number of proteins with changed abundance comparing red deer antler and sika deer antler make us to analyze the extent of shared and non-overlapping differences. A total of 578 proteins were identified in both the red deer antler proteome and the sika deer antler proteome, while 114 proteins were exclusively identified in sika deer antler (Fig. 1, Table S2). However, we found that only 19 exclusive proteins for red deer antler were observed (Fig. 1, Table S2). Among these antler proteins existed in both red deer and sika deer, 30 differentially expressed proteins (DEPs) were up-regulated and 17 down-regulated in the sika deer antler (Fig. 1, Table S2). Our current proteomics data indicated that more type of proteins were identified in sika deer antler than that in red deer.

Gene ontology annotation and enrichment analysis of differentially expressed proteins

To further study protein function in deer antler, Gene ontology (GO) annotation was processed to differentially expressed proteins by Blast2GO software. All terms were counted non-exclusively, when one protein possessed more than one term for biological process (BP), molecular function (MF), and cellular component (CC). The result of GO annotation was shown in Fig. 2 and Table S3. The most abundant category in molecular function was binding proteins (38%), especially ion bonding proteins (21%) and organic cyclic compound binding proteins (20%) (Fig. 2). The second highly abundant category in molecular function was catalytic activity proteins (23%), especially hydrolase activity proteins (10%). The cellular component analysis showed that the differentially expressed proteins were mainly located in the intracellular region, organelle and membrane (Fig. 2). 23 and 18% of proteins are in the organelle and membrane, respectively, whereas 9% of differentially expressed proteins are localized in the extracellular region. The major functional terms under biological process category were metabolic processes, cellular processes and biological regulation (18, 27, and 11%, respectively) (Fig. 2). The nitrogen compound metabolic process, which is a child term of metabolic process, possessed 22 differentially expressed proteins from two antler groups (Table S3).

Figure 1 Venn diagram of identified proteins from sika deer and red deer groups.

The identified antler proteins in at least two of three replicates were shown in the diagram. The overlapping regions indicate the number of shared proteins. The number above or below the horizontal line in each portion shows the number of up- and down-regulated shared proteins between sika deer group versus red deer group.

Figure 2 GO annotation for the differentially expressed antler proteins between sika deer and red deer.

The GO annotation results are displayed under three main categories: biological process, molecular function, and cellular component.

To reveal the biological difference between sika deer and red deer antler, GO enrichment analysis was performed for differentially expressed proteins on three categories (Fig. 3). We found that differentially expressed proteins were mainly located in clathrin-coated pit/vesicles, organelle, specially mitochondrion and cytoskeleton. On the molecular function category, a high abundance of DEPs was enriched into transporter activity and ATPase activity. DEPs were mainly involved in the regulation of gene expression and extracellular structure organization under biological process.

Figure 3 GO enrichment analysis for the differentially expressed antler proteins between sika deer and red deer.

The color of the bar chart represents the significance of the GO Enrichment. The color gradient corresponds to the magnitude of P value . The gradient becomes red, which indicates a low P value. The value on the top of the bar chart is the rich factor (rich factor ≤ 1). The rich factor is the ratio of the number of differentially expressed proteins associated with a certain GO term to the number of all identified proteins associated with a certain GO term.

KEGG annotation of differentially expressed proteins

KEGG analysis was used to reveal the functional pathways about differentially expressed proteins. A total of 180 DEPs were processed to obtain 163 KEGG pathways (Table S4). The top 20 pathway terms were shown in Fig. 4. These differentially expressed proteins were mainly classified into PI3K-Akt signaling pathway, thermogenesis, oxidative phosphorylation, ECM-receptor interaction and focal adhesion, ribosome, synaptic vesicle cycle, complement and coagulation cascades, and other categories related with disease.

Figure 4 KEGG annotation for the differentially expressed antler proteins between sika deer and red deer.

The top of the bar chart shows the number of identified proteins involved KEGG pathways.

Discussion

Proteins involved in oxidative phosphorylation

During antler regeneration stage, mesenchymal cells (MSCs) differentiate into chondrocytes participated in endochondral ossification, is one of the two key processes during bone development (Rucklidge et al., 1997). Previous study has shown that throughout the osteogenesis of MSCs, the copy number of mitochondrial DNA, oxygen consumption rate, OxPhos enzymes, mRNA associated with mitochondrial biogenesis, and intracellular ATP content were increased along with the significant decrease of endogenous reactive oxygen species (Chen et al., 2008; Pattappa et al., 2011). The production of ATP is mainly driven by OxPhos enzymes located in electron transport chain, where the proton gradient is built across the inner membrane by NADH-coenzyme Q oxidoreductase (complex I), Q-cytochrome c oxidoreductase (complex III), and cytochrome c oxidase (complex IV), which can drive ATP synthesis through ATP synthase (complex V) (DiMauro & Schon, 2003).

Table 1 Differentially expressed proteins involved in KEGG pathways.

UniProt ID	Protein Name	RD2/SD1	P-value	
Oxidative phosphorylation pathway	
A0A172ZQ52	ATP8	2.232388	0.018918192	
A0A212CA05	QCR1, UQCRC1	SD1a	–	
A0A212CD21	ATPeV1E, ATP6E	SD1a	–	
A0A212CE28	COX6C	SD1a	–	
A0A212D8C1	COX6A	SD1a	–	
A0A212D8Y9	QCR9, UCRC	SD1a	–	
A0A212DI17	NDUFS3	SD1a	–	
Ribosome pathway	
A0A212CGQ3	Large subunit ribosomal protein L14e	0.466	0.00337	
A0A212CIR2	Large subunit ribosomal protein L35A	0.429	0.0408	
A0A212CD44	Large subunit ribosomal protein L3e	SD1a	–	
A0A212CH34	Large subunit ribosomal protein L5e	SD1a	–	
A0A212CL29	Large subunit ribosomal protein L5e	SD1a	–	
A0A212CVE3	Large subunit ribosomal protein L4e	SD1a	–	
Extracellular matrix proteins	
A0A212C5S3	Laminin, alpha 4	0.400	0.0117	
A0A212C632	Laminin, alpha 3/5	SD1a	–	
A0A212C6B1	Collagen, type IX, alpha	SD1a	–	
A0A212CFD7	Collagen, type IX, alpha	SD1a	–	
A0A212D070	Cartilage oligomeric matrix protein	SD1a	–	
E7D7Z2	Osteopontin	SD1a	–	
PI3K/Akt pathway	
A0A212C5S3	Laminin, alpha 4	0.400	0.0117	
A0A212D1I8	Immunoglobulin heavy chain	RD2a	–	
A0A212CMX1	Serine/threonine-protein phosphatase 2A regulatory subunit B	RD2a	–	
A0A212C632	Laminin, alpha 3/5	SD1a	–	
A0A212C6B1	Collagen, type IX, alpha	SD1a	–	
A0A212CF73	Guanine nucleotide-binding protein subunit gamma-12	SD1a	–	
A0A212CFD7	Collagen, type IX, alpha	SD1a		
A0A212D070	Cartilage oligomeric matrix protein	SD1a		
A0A212DF15	HSP90A	SD1a		
E7D7Z2	Osteopontin	SD1a		
Notes.

a Proteins are expressed exclusively in a certain antler group.

According to KEGG annotation analysis, six up-regulated proteins in sika deer antler participated in oxidative phosphorylation (OxPhos) (Table 1). Of these six up-regulated proteins, vacuolar-type H-ATPase subunit E (ATP6E) belong to superfamily of related ATP synthases. Co-expression of the ATP8 and ATP6 genes provided stable protein expression for complex V assemble of the oxidative phosphorylation complexes (Boominathan et al., 2016). The up-regulation of the V-ATPase gene (ATP6V0D2) was found in the zone of hypertrophic chondrocytes, which was part of the process of endochondral ossification (Ayodele et al., 2017). The antlers were regenerated through a modified endochondral ossification process that involves the remodeling of cartilage (Faucheux et al., 2001). In this case, ATP6 could be the key enzyme involved in ATP production during antler development, whereas ATP8 may take over the roles of ATP6 in red deer. In addition, ATP8, which plays important roles in lipogenesis, was found during the stage of porcine adipose tissue development (Zhang et al., 2016). Weiss et al. (2012) reported that ATP8 mutation led to mitochondrial ROS generation with dysregulation of mitochondrial OxPhos system. It indicated that the high abundance of ATP8 in red deer might maintain the stable of mitochondrial OxPhos system when antler regeneration faced an increasing demand for energy. Moreover, cytochrome c oxidase subunit 6a (COX6A) and cytochrome c oxidase subunit 6b (COX6B) were significantly up-regulated in sika deer (Table 1). The enzyme cytochrome c oxidase (COX, also called complex IV), that consist of 14 subunits, is part of mitochondrial respiration oxidase (Balsa et al., 2012; Castresana et al., 1994). The three key subunits of COX are synthesized in mitochondrial to form the functional core of the enzyme, which is surrounded by 11 small nuclear-coded subunits (Balsa et al., 2012). The activity of cytochrome c oxidase, complex IV of the mitochondrial respiratory chain, has been shown to regulate the activity of osteoclasts necessary for adaptive bone remodeling (Gandhi et al., 2017; Miyazaki et al., 2003). The disruption of osteoclast COX activity could lead to the loss of bone-resorbing activity. The high expression of cytochrome c oxidase, accompanied with the up-regulation of ATP6 in sika deer, indicated that antler remodeling is more active than red deer. Accordingly, the higher ATP demand should be required for sika deer antler regeneration.

Previous studies have found that the overexpression of UQCRC1 enhanced complex III activity in mice, and its down-regulation was related with significant dysfunction in mitochondrion of epithelial cells (Kriaucionis et al., 2006; Shibanuma et al., 2011). This finding indicated that the up-regulation of UQCRC1 in sika deer (Table 1) contributes to maintain normal mitochondrial homeostasis, which is essential for osteoblast mediated bone formation (Pan et al., 2018). Etzler et al. (2017) have reported that the up-regulation of Complex III in HEK293 cells (human embryonic kidney cells) could enhance electron transport efficiency to survive from oxidative stress. Also, it was demonstrated that ROS inhibited the proliferation of chondrocytes and induced the chondrocyte hypertrophy (Morita et al., 2007). Therefore, the current proteomic study indicated that the antler development in sika deer require more complex III to regulate mitochondrial respiration in response to ROS, which was produced during endochondral ossification.

Recent work reported that the lower expression of NDUFS3 under oxidative stress impaired the Complex I assembly, which resulted in reduced Complex I content (Sen et al., 2015). Interestingly, the up-regulation of antioxidants, for example SOD2 and glutathione, could recover the complex I content (Gopal et al., 2018; Sen et al., 2015). According to the high abundance of NDUFS3, the content of antioxidants in sika deer antler should be higher that red deer (Table 1).

The high abundance of large subunit ribosomal proteins in sika deer

Another notable difference between red deer and sika deer antler was the higher abundance of large subunit ribosomal proteins (RPLs) in sika deer (Table 1). It is well known that ribosomes, as translational hub across the all living organisms, produce proteins based on mRNA template (De la Cruz, Karbstein & Woolford Jr, 2015; Khatter et al., 2015). The 80S ribosome in eukaryotes is comprised of a small (40S) and large (60S) subunit (Khatter et al., 2015). The up-regulation of ribosomal proteins was necessary to meet protein synthesis requirements in the rapid growth stage of coho salmon (Causey et al., 2018). It is consistent with our result, where the higher abundance of ribosomal proteins in sika deer as compared to red deer could lead to synthesize a greater variety of proteins in sika deer (Table S2). Therefore, ribosomal protein played a vital role in cell growth and proliferation by the regulation of protein biosynthesis. In bone marrow rRNAs, 60S subunit biogenesis associated with cycle circle (Khincha et al., 2016; Ruggero & Shimamura, 2014). The mutation of RPL35A contributed to Diamond-Blackfan anemia, that is an inherited bone marrow failure syndrome (Khincha et al., 2016). Moreover, RPL4 was identified as a crucial regulator involved in MDM2-p53 pathway (He et al., 2016). RPL4 directly binds to MDM2 could significantly inhibit the ubiquitination and degradation of p53, resulting in the up-regulation of p53. Interestingly, RPL5 and RPL11 also could bind to MDM2 in the same way for p53 induction under the assistant of RPL4 (Fumagalli et al., 2012; He et al., 2016; Sun, Dai & Lu, 2008). Further study had reported that the knockdown of large ribosome subunit proteins, RPL7a, RPL11, RPL5, RPL14, RPL26 and RPL35 would impair the ribosomal biogenesis (Bailly et al., 2016). The activation of p53 may promote endochondral ossification through interaction with proteins related with matrix mineralization and chondrocyte maturation or apoptosis (Li et al., 2012b). However, rpl3 regulated the cell cycle progression and apoptosis induction through p53-independent manner (Russo et al., 2016; Russo & Russo, 2017). The combination of the data suggested that high abundance of ribosomal protein large subunits in sika deer improved antler development, involving the process of endochondral ossification. Intriguingly, the content of ribosomal proteins in red deer was at relatively low level in contrast to sika deer.

Difference of extracellular matrix proteins

Type IX collagen (COL9) and cartilage oligomeric matrix protein (COMP), which are specifically expressed in cartilage tissue, participate in the regulation of cartilage development (Hecht et al., 2005; Myllyharju, 2014). The skeletal deficiencies of mutations in COL9 or COMP strongly suggested that an adequate cartilage matrix was essential for extracellular matrix assembly and homeostasis (Myllyharju, 2014; Shi et al., 2015). Mutations in COL9 or COMP have been shown to cause multiple epiphyseal dysplasia (MED), which was a rare genetic disorder (Briggs & Chapman, 2002; Briggs et al., 1995). collagen IX knockout mouse model have implied that reduction of collagen IX changed the time course of callus differentiation during fracture repair (Opolka et al., 2007). The maturation of cartilage matrix was delayed in collagen IX knockout mice during the repair phase and the content of cartilage matrix was at relatively low concentration (Opolka et al., 2007). In addition, collagen IX will be gradually reduced as the mineralization of cartilage tissue. In the case of mutations in COMP, it reduced not only COMP secretion but type IX collagen in cartilage matrix (Hecht et al., 2005). In our antler proteomics study, the content of COL9 and COMP in sika deer was much higher than red deer (Table 1), which indicated that cartilage matrix assemble in sika deer was more active and red deer may possess more mineralized tissue zone. Consistently, laminin alpha 4 (LAMA4), that are the major constituent of hypertrophic chondrocytes, was highly expressed in sika deer (Table 1). It has reported that LAMA4 was implicated in chondrocyte mobility and chondrocyte hypertrophic zone (Fuerst et al., 2011; Moazedi-Fuerst et al., 2016). LAMA4 binding with integrins improved the chondrocyte mobility to promote the antler development, whilst the interaction between LAMA4 and integrins activated PI3K-AKT pathway to regulate the proliferation, differentiation and survival of chondrocytes. Osteopontin (OPN) is an extracellular matrix glycoprotein involved in bone remodeling (Denhardt & Noda, 1998). OPN was found to be secreted by different cell types as diverse as macrophages, epithelial cells, and osteoclasts (Rittling, 2011; Tardelli et al., 2016). OPN, as migratory cytokine, stimulated the osteoclast migration to improve bone resorption. OPN is markedly expressed by MSCs and can be further up-regulated during the osteogenic differentiation (Chen et al., 2014; Rickard et al., 1994). OPN abundantly expressed in sika deer represented that the differentiation of MSCs putatively existed in the whole antler development stage. However, the abundance of OPN in red deer during late antler regeneration stage was significantly down-regulated, thereby retarded the mobility of osteoclast.

PI3K-Akt pathway

The Phosphoinositide 3-Kinase/Akt pathway is implicated in multiple cellular processes, including cell cycle entry, cell growth, cell survival and cell migration (Cantley, 2002; Liu et al., 2009). A set of hormones, cytokines, and extracellular matrix (ECM) proteins, as stimulators, can activate PI3K/Akt pathway (Liu et al., 2009). Furthermore, activation of Akt signaling in cartilage development enhanced the proliferation of chondrocytes and suppressed the hypertrophic differentiation of chondrocytes through the down-regulation of Runx2 expression, which participated in chondrocyte terminal differentiation (Kita et al., 2008). Compared with red deer, four ECM proteins in sika deer antler were significantly up-regulated (Table 1). Thereby, ECM proteins interacting with integrins can activate PI3K/Akt pathway through the action of focal adhesion kinase (FAK) in the sika deer. Meanwhile, the high abundance of guanine nucleotide-binding protein subunit gamma-12 (GNG12) in sika deer could can improve antler regeneration by activating PI3K (Table 1) (Luo et al., 2018). Another up-regulated protein belonging to PI3K/Akt pathway in sika deer, heat shock protein 90 (HSP90A), may contribute to mesenchymal stem cell migration to facilitate the antler development (Table 1) (Gao et al., 2015). Conversely, the up-regulation of serine/threonine protein phosphatase 2A (PP2A) in red deer will inhibit the dephosphorylate of Akt to inhibit PI3K/Akt pathway (Table 1) (Eichhorn, Creyghton & Bernards, 2009).

Conclusions

Our comparative proteomics study provided the comprehensive analysis of the differentially expressed antler proteins between sika deer and red deer based on label-free quantitative proteomics. The protein species and their abundance of sika deer antler were higher than in red deer. Differentially expressed proteins during the antler regeneration stage mainly participated in the regulation of several pathways including oxidative phosphorylation, ribosome, extracellular matrix interaction, and PI3K-Akt pathway.

Supplemental Information

Table S1 The proteins identified in both sika deer (SD1) and red deer (RD) antler

Click here for additional data file.

Table S2 Differentially expressed proteins between sika deer and red deer

Click here for additional data file.

Table S3 GO analysis for differentially expressed proteins between sika deer and red deer

Click here for additional data file.

Table S4 KEGG annotation for differentially expressed proteins between sika deer and red deer antler

Click here for additional data file.

Additional Information and Declarations

Competing Interests

Author Contributions

Animal Ethics

Field Study Permissions

Data Availability

The authors declare there are no competing interests.

Hang Su conceived and designed the experiments, performed the experiments, analyzed the data, contributed reagents/materials/analysis tools, prepared figures and/or tables, authored or reviewed drafts of the paper, approved the final draft.

Xiaolei Tang performed the experiments, approved the final draft.

Xiaocui Zhang contributed reagents/materials/analysis tools, approved the final draft.

Li Liu contributed reagents/materials/analysis tools, prepared figures and/or tables, approved the final draft.

Li Jing analyzed the data, contributed reagents/materials/analysis tools, prepared figures and/or tables, approved the final draft.

Daian Pan and Weijie Sun contributed reagents/materials/analysis tools, approved the final draft.

Huinan He analyzed the data, contributed reagents/materials/analysis tools, approved the final draft.

Chonghui Yang contributed reagents/materials/analysis tools, prepared figures and/or tables, approved the final draft.

Daqing Zhao approved the final draft, technical support.

He Zhang conceived and designed the experiments, prepared figures and/or tables, approved the final draft.

Bin Qi conceived and designed the experiments, authored or reviewed drafts of the paper, approved the final draft.

The following information was supplied relating to ethical approvals (i.e., approving body and any reference numbers):

The Animal Ethics Committee of Changchun University of Chinese Medicine provided full approval for this research.

The following information was supplied relating to field study approvals (i.e., approving body and any reference numbers):

Field experiments were approved by the Research Council of Changchun University of Chinese Medicine (project number: 17.03.001; Research Council Approval number:2017-120798).

The following information was supplied regarding data availability: PRIDE Archive: https://www.ebi.ac.uk/pride/archive/projects/PXD012090.

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
