# Peer review of "Comparative proteomics analysis reveals the difference during antler regeneration stage between red deer and sika deer"

_PeerJ, doi:10.7717/peerj.7299_

## Round 0.1 · original submission · Major Revisions

Your manuscript has been reviewed by two expert reviewers. Both reviewers found that the aim of the study is rather vague and therefore the experimental design and the conclusions are also questionable. The major problem is that the manuscript compares the proteome from regenerating antlers of two deer species, red deer (Cervus elaphus) and sika deer (Cervus nippon), but it is not specified whether the differences observed are indeed connected to antler morphological and regeneration differences between the two deer species, or they rather reflect differences in the timing of the regeneration process. The introduction lacks important background information related to antler development and the conclusions are not appropriately connected to the original question addressed. The reviewers also raised substantial criticisms related to the usage of bioinformatic, statistical and other methods. The authors did not define why RD2 and SD1 specimens were chosen for more detailed comparison.

A substantial revision of the manuscript is needed, including the Introduction, Materials and Methods, Results and Discussion. The authors should clearly describe which part of the antler they sampled and why, e.g. how far the saggittal sections protruded from the tip of the antler. Histological sections should be shown to verify that the samples of the two deer species were taken from comparable antler regeneration stages. A more detailed description and reevaluation of the bioinformatic and statistical data is also needed. Verification of the most important findings by independent methods is suggested to increase the quality of the work. All the tables suggested to be merged into one with 4 categories. Grammatical errors, typos and inconsistencies (e.g. names of corresponding authors) should also be corrected.

I suggest considering all the thoughtful and important comments of the reviewers and revising your manuscript in the light of their criticisms. Please provide your point-by-point response to the reviewers’ comments, when submitting your revised manuscript. Please explain clearly, if you do not agree with any of their comments or suggestions.

·

Basic reporting

See the attached pdf file

Experimental design

See the attached pdf file

Validity of the findings

See the attached pdf file

Additional comments

See the attached pdf file

Reviewer 2 ·

Basic reporting

1. English has to be improved in terms of grammar, syntax and typos and the language has to be edited by a professional/native speaker.

2. Introduction and background are relatively poorly covered. Some key references for antler gene expression are missing (Borsy et al, Mol Genet Genomics (2009) 281:301–313; Molnár et al, Mol Genet Genomics (2007) 277:237–248; Stéger et al, Mol Genet Genomics (2010) 284:273–287). A more detailed description of the antler development, as per the missing references, would be required. I think that the miRNA part of the introduction is irrelevant and should be omitted.

3. Structure of the manuscript complies with the criteria of Peer J, although acknowledgement is missing.

4. Figures are fine, but my comment on the language is also valid for them.

5. Raw data were supplied.

Experimental design

1. The study is within the aim and scope of the journal

2. The research question is not very well defined, it is a sort of much generalised statement about “distinctive differences of antler regeneration between sika deer and red deer” (line 62) and “to study the antler systems biology” (line 63). Antler system (?) biology is a very broad area and the authors did not provide any particular details what they would like to know by the study.

3. The investigation is not very well designed. Antler regeneration is a long, about 110-120 days, and we only know that sampling was done on day 80. We even do not know, because the precise description of the sampling is not given, that which exact part of the developing antler was sampled. I could only guess that samples were taken from a calcified and ossificated section of the antler. Since previous gene expression studies for both red deer (see missing references) and sika deer (Yao et al.) used the growing tip of the antler for sampling, the authors should define that which part of the antler they sampled and why.

4. Materials and methods. The precise tissue sampling (lines 75-84) should be described. Line 91: which lysates were combined? Line 93: full name for BCA should be given. Lines 128-130: settings for the BLAST search and GO analyses should be given. Lines 136-138: I am not certain why the authors used the Fisher’s exact test for the enrichment analysis. Enrichment tests typically require a list of gene terms (name or similar), which is then compared to a background list, while Fisher’s exact test requires numerical data. The authors should give a detailed explanation on this and/or repeat the analysis by the GSEA module of BLAST2GO or BINGO or similar.

Validity of the findings

The authors have identified a number of differentially expressed proteins in their study. A number of these (47) were common between red and sika deer, while others (19 and 114, respectively) were unique for the two species.

Regarding their results.

1. Samples are not paired, i.e. animal RD2-1 is not the pair of SD1-1. Therefore, calculating the log2 ratio between these two individuals, and between the other animals too, is meaningless, and thus the showing the heat map in Figure2 is unnecessary. RD2-1 could be paired with either SD1-2 or SD1-3 too and then log2 expression ratio would be different. Theoretically, nine combinations could be generated, each resulting in different ratios. I suggest to eliminate Figure2. The averages and their ratio as shown in Table S2 is sufficient.

2. In Table S2, the animal IDs are missing in columns J, K, and N, O. 3.

3. In total, the authors identified 1060 proteins were identified (line 143), but for 82 of these there is no LFQ intensity in TableS1, and I could not figure out why those were listed. Regarding this, column headings need a short definition to show for example in a separate additional file listing all Supplementary tables. So, 978 proteins had LFQ intensity and it needs an explanation how this number was reduced to 578+114+19 = 711. I can guess that for example a protein which was detected only in one red deer was not considered as unique, but still it should be describe either in the method or in text. Moreover in Table S3, there are 861 protein are listed on the “enrichment” sheet, but the sum of the protein in the three ontology group is 864. There is a discrepancy in these numbers and the authors should investigate, what went wrong. So how this 861 number came?

4. I am curious, why the enrichment study was performed on all genes, whatever number. Enrichment analysis on all genes has no meaning in term of the differences between the two species. I suggest to redo the analysis on the 578 common and the 114 and 19 unique genes in order to see what sort of functional differences are between red and sika deer.

5. The enrichment data looks strange. If you perform an analysis whose output is both P-value and FDR, then if the analysis was sound, the P-value is smaller than FDR by about 1, 2 or more order of magnitude. For example, a P-value of 1.40E-09 should correspond to and FDR of 1.00E-07, 0.025 to 0.055 and 0.78 to 0.85 (not significant). But a 0.002557 P-value with a 0.731 FDR indicates that something is wrong with the analysis. Also, different P-values should not have the same FDR. Regarding this, what is the “rich factor” in Figure 4 and Table S3?

6. Talking about up- and downregulation can be tricky. The authors identified 30 proteins upregulated in sika (obviously downregulated in red deer) and 17 proteins downregulated in sika (upregulated in red deer). In the text (lines 154-160) they are mentioning the ratios from the viewpoint of sika (“30 up-regulated, 17 down-regulated in sika deer group”). However in Table S2 they show the red/sika ratio. I think it would be better to show the sika/red ratio, which would reflect better to their preferred way of talking about ratios relative to sika. In Table S2 it would be also nice to show the log2 ratio of the expression ratio, which would very clearly show the 2-fold change threshold (+1 for 2-fold upregulation, -1 for 2-fold downregulation), this is the most common way of displaying expression ratios anyway.

7. In the discussion, the authors cite a number of publications related to bone development and discuss the differentially expressed they identified in that contest. They also use physiological result from two publications (Chen et al. 2008 and Pattappa et al. 2011) as their focal point for their discussion. These two publications are about human mesenchymal stem cells. But they do not use the Li et al. 2012a publication and the above mentioned red deer studies to discuss their results. One major source of the problems with the discussion that we do not precisely that which portion of the antler was used for sampling. If the site of sampling was a fully differentiated bone part of the antler, then all of the bone development literatures are not very valid. For me it seems to be that the authors mined for literatures which describes those genes that they identified and are mentioned in relationship with bones, but the two is not necessary means causality. It is difficult to explain anyway that why a gene related to bone development in differentially expressed between tow deer species in an established tissue, if so. In the case of the authors if would reconsider the discussion strategy and mainly focus on antler development studies.

---

## Round 0.2 · Major Revisions

The authors have revised their manuscript by making some modifications in the text and Figures, but it seems that they have not understood the most important comments of the Reviewers and the Editor. The main problem is that the aim of the study is not clear and therefore the experimental design and the conclusions are questionable. It is still not specified whether the differences observed are indeed connected to antler morphological and regeneration differences between the two deer species, or they rather reflect differences in the timing of the regeneration process. As I recommended earlier it would be important to show histological staining of sections used for proteomic analyses thereby verifying that the samples of the two deer species were taken from comparable antler regeneration zones. Thus comparing the proteome of the subsequent differentiation zones between the two deer species could lead to conclusive results and meaningful discussion.

The manuscript still needs a substantial revision including the proteomic and statistical analyses. The conclusions should be connected to the aim of the study, should be limited to those supported by the results and should be clearly stated.

As pointed out by Reviewer 2, there are still serious grammatical and typing errors in the manuscript, which should be corrected.

I suggest revising your manuscript by considering the comments of the reviewers.

[]

Reviewer 2 ·

Basic reporting

English still needs a serious editing. I do not think that it was checked properly. For example, the words proteome, proteomic and proteomics are used in an incorrect way. The articles “The” and “a” are used incorrectly. Proteomics is the science, proteome is the sum of proteins. The authors sometimes changed the good word to bad, and Many other syntax and wording errors. A typical example: “more kinds of proteins…”. This should be “more proteins” or “more type of proteins. Typing errors are present too.

If a reviewer ask for some information, it not only means that the authors should answer it, but it also should go into the revised text. A typical example is the question from Reviewer 1 “What software was used for the differential analysis? What kind of test was used (ANOVA, t-test... )? Were protein abundances normalized? The answer was not added to the manuscript. It should be.

Experimental design

Line 95: sagittal means: “situated in the direction of the sagittal suture; said of an anteroposterior plane or section parallel to the median plane of the body.” (From the Free Dictionary). From this it is difficult to find out how the antler tip was sectioned. I guess that the authors made cross section of the antler. Why don’t you say so?


In the methods section, line 104, it is still not clear that what the combination of lysates are. Be more precise.

Sampling. A 5 cm piece of the antler tip was removed and then sectioned into 5 to 8 mm sections. That means 6 to 10 sections. They were combined for protein extraction or extracted separately and then extractions were combined?

Validity of the findings

I still have serious problems with data presentation.
1. Despite that the corresponding author wrote so, the headings for some of the intensities in Table S2 were not corrected (animal ID number).

2. There is no legends for the supplementary tables. Thus it is difficult to understand them. The authors should briefly but precisely describe what the headings are. What is “coverage” for example, and how it was calculated. Let see an example: In the case of protein A0A220IGB1, there are four protein in the group. Then the 17 peptides were identified in all four proteins, or just the 8 unique peptides were identified in all four proteins? 17 peptides were identified in each sample or they were identified in the six samples? The peak intensity is the sum of the intensity of all peptides in a sample? I am not sure that the peptide numbers should be given in any table.

3. DO NOT show those proteins in Table S1 that were eliminated from further analysis. There is no reason to show them. But precisely describe in the main text what data were eliminated and why, but do not show them!

4. Have a legend for all supplementary tables.

5. In line 163 the authors say that “In total, 4721 unique peptides and 1060 proteins were identified”. But in Table S1, the total number of unique peptides is 4439. Why is that? The total number of peptides on the other hand is 5459.

6. In Table S1, the peptide numbers are a bit confusing. If there are “2” in Peptide cell, it does not mean that two peptides were present in the sample, it means that two different peptide sequences were identified, their absolute number is nor know. From the intensity data we just can predict that
2-fold higher intensity (2-fold upregulation) in one species that the absolute number of peptides were roughly 2-fold higher in one samples then in the other, but it can be 2 and 4 or 25 and 50. This is another reason to provide Table legends.

7. Figure 2 is still problematic. The scale looks for me log2. So it must be expression ration, because the intensity values in Table S1 cannot produce negative numbers. But there are six columns on the figure the same number as the samples. If you calculated rations and paired the samples then you should have 3 columns or 9 if you pair each samples. Moreover, you did not provided log2 values for the ratios in Table S1, so where the numbers for colour scale for the Heatmap was obtained from? I cannot accept your answer for this suggestion. If I suggested: remove it, then I meant REMOVE it. You could not defend, why the figure should stay.

8. For Table S3 give the log2 values of the ratios, and explain what the “t test p values” is. If it is for the averages then that column should go before the ratio column.

9. You did the enrichment analysis on the 180 DEGs. That is fine, but I suggested that do it (and I mean do it), on the common and the specific genes, because this is the only way to find species specific and common functions. So use the 114, 19 and the 47 genes, and the last group would be even better to separate into up- and down regulated groups. Your answer regarding this talking about the background gene list construction, did not answer my comment.
You used only the 1060 protein identified in you experiments as a background list for the enrichment analysis. This is ridiculously low number and, therefore, your analysis is very limited and far from acceptable. In an enrichment analysis, you have to use the widest possible gene set of the organism as a background, not only an organ specific or limited gene set. The UniProt database has 21,509 Cervus protein entries, so for the new analysis, which I require, you have to download this entries and use those proteins as background that have a GO annotation.

---

## Round 0.3 · accepted · Accept

Your revised manuscript has been reviewed by two expert reviewers. Both reviewers found your work suitable for publication in PeerJ.

·

Basic reporting

cf General comments

Experimental design

cf General comments

Validity of the findings

cf General comments

Additional comments

The authors addressed all the comments in my opinion.

Reviewer 2 ·

Basic reporting

Questions were answered

Experimental design

Questions were answered

Validity of the findings

Questions were answered